# Score-based Explainability for Graph Representations

**Ehsan Hajiramezanali**                                           *hajiramezanali.ehsan@gene.com*
*Genentech*

**Sepideh Maleki**                                                  *maleki.sepideh@gene.com*
*Genentech*

**Max W Shen**                                                       *shen.max@gene.com*
*Genentech*

**Kangway V. Chuang**                                               *chuang.kangwat@gene.com*
*Genentech*

**Tommaso Biancalani**                                        *biancalani.tommaso@gene.com*
*Genentech*

**Gabriele Scalia**                                               *scalica.gabriele@gene.com*
*Genentech*

**Reviewed on OpenReview:** *https://openreview.net/forum?id=K6DKrrpYpJ*

## Abstract

Despite the widespread use of unsupervised Graph Neural Networks (GNNs), their post-hoc explainability remains underexplored. Current graph explanation methods typically focus on explaining a single dimension of the final output. However, unsupervised and self-supervised GNNs produce d-dimensional representation vectors whose individual elements lack clear, disentangled semantic meaning. To tackle this issue, we draw inspiration from the success of score-based graph explainers in supervised GNNs and propose a novel framework, grXAI, for graph representation explainability. grXAI generalizes existing score-based graph explainers to identify the subgraph most responsible for constructing the latent representation of the input graph. This framework can be easily and efficiently implemented as a wrapper around existing methods, enabling the explanation of graph representations through connected subgraphs, which are more human-intelligible. Extensive qualitative and quantitative experiments demonstrate grXAI's strong ability to identify subgraphs that effectively explain learned graph representations across various unsupervised tasks and learning algorithms.

## 1 Introduction

Graph neural networks (GNNs) have become increasingly useful in a wide range of applications, such as drug discovery (Lu et al., 2024; Hajiramezanali et al., 2020), large-scale social networks (Hajiramezanali et al., 2019), and recommender systems (Hasanzadeh et al., 2019). However, due to their complexity and opacity, understanding GNNs can be challenging for human users. To tackle this issue, several tools have been developed to explain supervised GNNs in terms of their predictions. In the supervised setting, a GNN model learns to map input graphs to labels ($f(\cdot) : \mathcal{G} \longrightarrow \mathcal{Y}$), and explanations shed light on the model's prediction of a specific label. The interpretability of *model-agnostic* GNN explanation methods stems from the fact that the particular label[1] of interest, such as a chemical property, is meaningful to humans (Figure 1, left).

---

[1]The term label used in the context of explainability refers to the predicted label for the test input graph by the GNN model being explained, which is distinct from the ground truth label used in the context of supervised ML (Ying et al., 2019).

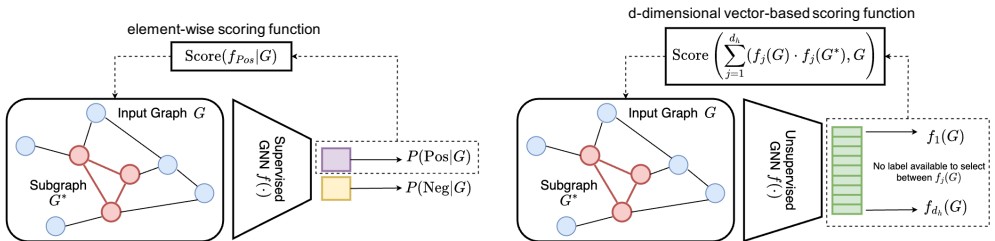

Figure 1: **(Left)** Post-hoc graph explainability methods typically provide explanations based on a specific label of interest. **(Right)** Our framework for explaining graph representations provides explanations based on a d-dimensional representation vector. The *Score* refers to a range of post-hoc explanation functions proposed in the literature, including Shapley and Saliency.

On the other hand, unsupervised GNNs map input graphs to representations ($f(\cdot) : \mathcal{G} \longrightarrow \mathcal{H}$) that cannot be easily interpreted using existing supervised explanation methods. Although these methods can be used to understand individual elements in the representation space ($h_i \in \mathcal{H}$), such explanations are not meaningful to humans unless the elements have a natural semantic meaning (Lin et al., 2023). Unfortunately, the meaning of individual elements in the representation space is generally unknown. Two possible solutions have been proposed: the first approach enforces semantic meaning in the representations, but it requires concept labels for every training sample, which is often impractical (Koh et al., 2020). The second approach enforces disentangled representations and manually identifies semantically meaningful elements (Paige et al., 2017), but this approach is not model-agnostic and requires potentially unwanted changes to the modeling and training process (Lin et al., 2023).

Another possible approach to solve the aforementioned issue may involve utilizing the existing methods to explain each element of the representation vector independently and then averaging over them (Crabbé & van der Schaar, 2022). However, this solution is not sufficient to provide a global understanding of the latent representation as a whole and does not necessarily generate connected subgraphs as explanations, which are important for human understanding and to improve consistency (Hajiramezanali et al., 2023). Additionally, many existing post-hoc explanation methods, such as gradient- or perturbation-based methods, rely on multiple perturbations or backpropagations of the GNN model to generate explanations for each input graph, making them time-consuming and impractical when applied to each element of high-dimensional representations (Chuang et al., 2023). For instance, Shapley-based graph explainers that utilize the GNN architecture to approximate Shapley values are computationally expensive (Xie et al., 2022). This cost escalates linearly with the dimension of the representation vector, rendering them impractical for explaining unsupervised GNNs. Addressing this issue would require non-trivial extensions.

**Our main contributions are:** (i) We introduce a general framework, grXAI, that adapts model-agnostic *score-based* graph explainability methods to explain (representation) vectors. We do this by defining an auxiliary cosine similarity function as a wrapper around the unsupervised GNN to interpret. (ii) We apply our framework to different types of graph explainability approaches, including SubgraphX, and introduce the first graph explainability method for unsupervised settings that can explain the model using *connected subgraphs.* (iii) While motivated by the explanation of purely unsupervised graph representation vectors, the proposed approach is beneficial even for supervised multi-class classifications, generating more robust and accurate explanations for high-uncertainty cases. (iv) Our experiments show that our methods achieve state-of-the-art results, both qualitatively and quantitatively, on various self-supervised tasks, including graph representation and node embedding.

**Motivation.** Learning meaningful representations of graph-structured data with GNNs is important in many fields, particularly when labeled data are scarce due to costly and time-consuming experiments(Hasanzadeh et al., 2021). To this end, applications of GNNs in a wide range of domains, such as drug discovery and molecular biology (Hasanzadeh et al., 2022), have inspired recent unsupervised and self-supervised strategies to learn from massive corpora of unlabeled structured data (Sun et al., 2019; You et al., 2020; Hu et al., 2019). The ability to explain unsupervised GNNs can provide valuable insights into the learned representation,

Table 1: Comparison of different graph explainers.

| | Explainer | Supervised GNNs | Unsupervised GNNs | d-dimensional vector | Connected subgraphs |
|---|---|---|---|---|---|
| **MI-based** | GNNExplainer (Ying et al., 2019) | ✓ | | | |
| | PGExplainer (Luo et al., 2020) | ✓ | | ✓† | |
| | TAGE (Xie et al., 2022) | ✓ | ✓ | ✓ | |
| **Score-based** | IG & Saliency (Simonyan et al., 2013; Sundararajan et al., 2017) | ✓ | | | |
| | SubgraphX (Yuan et al., 2021) | ✓ | | | ✓ |
| | **grXAI (ours)** | ✓ | ✓ | ✓ | ✓ |

† This method is designed to handle probability vectors, such as the outputs of a softmax layer, and is not capable of handling embedding vectors learned through unsupervised settings.

help researchers understand and compare graph representation learning methods, as well as assist users in effectively monitoring and debugging these models during deployment. Our proposed framework enables score-based explainability in unsupervised settings in a scalable way, making it feasible and effective for real-world applications. Furthermore, our framework offers benefits even in supervised settings, proving particularly helpful when the model's predictions are uncertain.

## 2 Related work

With many recent advances in GNNs and their numerous applications across different fields, explainability methods have become critical for providing insight into their predictions. To this end, many approaches have been proposed to explain the predictions of *supervised* GNN models (Baldassarre & Azizpour, 2019; Ying et al., 2019; Luo et al., 2020; Xie et al., 2022; Yuan et al., 2021; Zhang et al., 2022; Ye et al., 2024; Huang et al., 2024; Hasanzadeh et al., 2020).

These methods can be divided into four main categories (Yuan et al., 2020): gradient-, perturbation-, decomposition-, and surrogate-based methods. In this paper, we mainly focus on gradient-based and perturbation-based methods. The gradient-based methods are generally fast and easy to implement, but they may not be able to capture more complex relationships in the data (Yuan et al., 2021; 2020). On the other hand, perturbation-based methods can be more computationally expensive, but they generally achieve state-of-the-art performance in terms of explanation quality (Xie et al., 2022). The common characteristic of most of these methods is that they require labels to specify which element of the GNN's output to explain.

Instead of explaining individual elements in the representation, recent approaches have focused on explaining multi-dimensional representation vectors as a whole, which is critical to explaining unsupervised models. These include TAGE (Xie et al., 2022) in the context of GNNs, and methods such as COCOA (Lin et al., 2023), RELAX (Wickstrøm et al., 2023), Label-Free XAI (Crabbé & van der Schaar, 2022), and CoRTX (Chuang et al., 2023) for other domains.

TAGE (Xie et al., 2022) successfully addresses *unsupervised* GNN settings using a contrastive loss approach. However, it has two main limitations (see Table 1). First, TAGE is a mutual information-based (MI-based) explainability method; however, it has been shown that score-based explainability methods outperform MI-based ones in supervised GNN settings (Yuan et al., 2020). Hence, an open problem is how to leverage existing score-based methods to explain GNNs in unsupervised settings. Second, TAGE focuses on explaining the importance of graph edges, but ignores the substructures of graphs. However, explanations consisting of connected subgraphs have not only been deemed more intuitive to humans in practice (Yuan et al., 2021; 2020), but they are also more consistent in supervised GNN settings (Hajiramezanali et al., 2023).

While the explainability of graph representations remains largely unexplored, several methods have been proposed to explain unsupervised models for unstructured data. The RELAX and LFXAI approaches share the goal of identifying features in the sample being explained (i.e., the explicand) that, if removed, would cause the altered representation to diverge from the original representation of the explicand (Lin et al., 2023). Chuang et al. (2023) introduce the contrastive real-time explanation (CoRTX) framework, which utilizes contrastive learning techniques. Unlike previous feature-based methods, Lin et al. (2023) introduce contrastive corpus attribution (COCOA), which allows users to choose corpus and foil samples in order to

identify features that make the explicand's representation similar to the selected corpus, but dissimilar to the foil samples. These methods do not directly generalize explainability methods designed for GNNs, which often take into account the non-Euclidean nature of the data, arbitrary size, and complex topological structure. Furthermore, the large number of possible subgraphs makes it more difficult to explain GNNs than their counterparts in Convolutional Neural Networks (CNNs). In contrast to these works and in line with Crabbé & van der Schaar (2022), our framework focuses on extending and generalizing existing graph explainability methods (which account for GNN-specific features) to the unsupervised representation learning case.

## 3 Preliminary

Our framework can generalize most of the existing score-based explainability methods to an unsupervised setting. Therefore, we first outline *supervised* score-based graph explainability and then overview two types as proof-of-concept examples. We will show how our framework can generalize them to explain (unsupervised) graph representations in Section 4. Specifically, we will discuss SubgraphX, which is currently the state-of-the-art perturbation-based model (Yuan et al., 2020; Hajiramezanali et al., 2023), followed by two computationally efficient gradient-based methods (Integrated Gradients and Saliency). Additional preliminaries less related to the proposed method, including graph self-supervised learning and TAGE, can be found in Appendix B.

**Notation.** Let $G = (\mathcal{V}, \mathcal{E})$ denote a graph on nodes $\mathcal{V}$ and edges $\mathcal{E}$ with the adjacency matrix $\mathbf{A}$ and $M$-dimensional node attributes $\mathbf{X} \in \mathbb{R}^{N \times M}$, where $N = |\mathcal{V}|$. We are given a trained GNN model $f(\cdot)$, which is optimized on all graphs (or nodes) in the training set and is then used for predictions. For graph classification, $f(\cdot) : \mathcal{G} \longrightarrow \mathcal{Y}$ maps each input graph $G \in \mathcal{G}$ to an output $\mathbf{y}_G = f(G) \in \mathcal{Y}$. For node classification, $f(\cdot) : G \longrightarrow \mathcal{Y}$ maps each input node $v \in G$ to an output $\mathbf{y}_v = f(v) \in \mathcal{Y}$. Please note that $\mathcal{Y} \subset \mathbb{R}^{d_y}$ and $d_y > 1$.

### 3.1 Score-based graph explainability in supervised settings

Given a pre-trained GNN model $f(\cdot) : \mathcal{G} \longrightarrow \mathcal{Y}$, and assuming $\{G^{(i)}\}_{i=1}^n$ is the set of subgraphs of the input graph $G \in \mathcal{G}$, where $n$ is the total number of subgraphs, these methods assign a *score* to each subgraph $G^{(i)} \in G$ based on its importance to the GNN prediction $f_j(G)$. Here, $f_j(G)$ denotes the $j$-th element of the GNN output. Intuitively, the score functions aim to identify the most important subgraphs $(G^*)$ that, when removed, decrease the predicted probability of a class of interest the most. Therefore, the main component of such models is the *scoring function* $\text{Score}(f_j(\cdot), G, G^{(i)})$, which calculates the importance of each possible subgraph with respect to a specific label of interest (Figure 1, left).

A straightforward way to obtain $G^*$ is to enumerate all possible subgraphs in $G$ (i.e. $\{G^{(i)}\}_{i=1}^n$), calculate their scores using the scoring function, and select the most important one as the explanation. However, this brute-force approach is an intractable combinatorial problem, as the number of potential candidates increases exponentially. Different approaches address this issue in various ways. Some methods incorporate search algorithms to efficiently explore the (connected) subgraphs, as seen in SubgraphX, while others calculate the importance of each node separately, similar to the gradient-based methods. The latter approach leads to disconnected subgraphs.

### 3.2 SubgraphX

Key properties of graphs can often be attributed to important and localized structural information. The goal of SubgraphX (Yuan et al., 2021) is to identify the most important *connected subgraph* within the input graph that contributes to the GNN's predictions. To achieve this, SubgraphX uses a scoring function to evaluate the importance of different subgraphs based on their interactions with the trained GNN.

Consider the set of connected subgraphs of $G$ as $\{G^{(i)}\}_{i=1}^n$, where $n$ is the total number of connected subgraphs. SubgraphX explains the GNN prediction $\mathbf{y}_j$ for the input graph $G$ as:

$$G^* = \underset{|G^{(i)}| \leq N_{\min}}{\arg\max} \ \text{Score}_{\text{Shapley}}(f_j(\cdot), G, G^{(i)}), \tag{1}$$

where $f_j(.)$ denotes the $j$-th element of the GNN output, $N_{\min}$ is an upper bound on the size of a subgraph, and the Shapley value (Kuhn & Tucker, 1953) has been used as the scoring function. To efficiently explore the space of possible connected subgraphs, SubgraphX employs Monte Carlo Tree Search (MCTS) (Jin et al., 2020; Silver et al., 2017) to guide the search process. Specifically, it builds a search tree in which the root is associated with the input graph and each of the other nodes corresponds to a connected subgraph. Each edge in the search tree denotes that the graph associated with a child node can be obtained by performing node-pruning from the graph associated with its parent node (Yuan et al., 2021). Please note that $N_{\min}$ in equation 1 is a hyperparameter of MCTS, which controls the size of the important subgraphs.

### 3.3 Gradient-based methods

Gradient-based methods, including Integrated Gradients (Sundararajan et al., 2017) and Saliency (Simonyan et al., 2013), address the problem of attributing the prediction of a black-box, here a GNN, to its input features. Saliency is a simple approach for computing input attribution, returning the gradient of the output with respect to the input features. This approach can be understood as taking a first-order Taylor expansion of the network at the input, and the gradients are simply the coefficients of each feature in the linear representation of the model (Simonyan et al., 2013). The value of these coefficients can be interpreted as the importance of the features (nodes/edges) in explaining the output of the black-box model. Integrated Gradients (IG) (Sundararajan et al., 2017) computes the integral of the gradients with respect to the inputs along the path from a given vector. Similar to Saliency, an attribution at input $\mathbf{x}$ relative to a baseline vector $\mathbf{x}_b$ represents the contribution of each individual input feature (nodes/edges) to the model prediction.

Formally, given an input graph $G$ and a GNN classifier $f(\cdot)$, these methods rank the nodes $v_i \in G$ based on their influence on the predicted label $f_j(G)$. Specifically, the importance of each node $v_i$ to the GNN output can be written as $\text{Score}(f_j(\cdot), G, v_i)$, where $\{v_i\}_{i=1}^{|\mathcal{V}|}$ are the nodes of $G$. We can then explain a GNN prediction based on a *disconnected* subgraph using the top-ranked nodes as $G^* = \{v_s\}_{s=1}^{\mathcal{V}_*}$. Here, $\mathcal{V}_*$ is the total number of important nodes.

## 4 Method

The grXAI framework is designed to provide explainability for unsupervised GNN models by identifying important subgraphs that can explain the d-dimensional representation vector generated by these models. However, current graph explainability methods have limited capability in handling d-dimensional embeddings, as their score functions are only applicable to a single dimension of the output (Xie et al., 2022). This limitation makes it challenging to determine which models' output(s) to interpret in unsupervised settings, as representation dimensions do not generally correspond to any meaningful quantity. To overcome this limitation, we first outline score-based graph explanations for d-dimensional outputs of GNNs in a supervised setting, gaining insights that will enable us to extend the framework to unsupervised GNN settings.

### 4.1 Explaining d-dimensional GNNs with score-based methods

Score-based graph explainability methods aim to explain the predictions of GNN models by computing an importance score $\text{Score}(f(\cdot), G, G^{(i)})$ for each subgraph $G^{(i)}$ of $G$. However, as mentioned in Section 3, existing importance score functions are designed to be applied to individual elements of the softmax output, represented by $f_j(.) \in \mathbb{R}$ for some $j \in 1, \ldots, d_y$, and cannot handle high-dimensional probability vectors. As a result, they approximate $\overline{\text{Score}}(f(\cdot), G, G^{(i)})$ as $\text{Score}(f_j(\cdot), G, G^{(i)})$, where $j = \arg\max_{k \in 1, \ldots, d_y} f_k(\cdot)$. That is, the importance score is solely based on the predicted class with the highest probability.

This approach limits the comprehensiveness of the graph explanation. Indeed, it does not account for the inherent uncertainty in the model's predictions, particularly in multi-class problems. Thus, it hinders a complete understanding of the model's behavior. One possible solution to address this issue is to compute the importance scores for each output component separately and then combine them into a weighted sum

across all components, using the associated class probabilities as weights, as follows:

$$\overline{\text{Score}}_{\text{Naive}}(f(\cdot), G, G^{(i)}) = \sum_{k=1}^{d_y} f_k(G) \cdot \text{Score}(f_k(\cdot), G, G^{(i)}). \tag{2}$$

This formulation considers the contribution of each class and is therefore accurate when class probabilities are more balanced, unlike the standard definition, which is only accurate for low-uncertainty predictions (i.e., when $f_j(G) \approx 1$). However, using equation 2 has important limitations. Primarily, searching for (connected) important subgraphs can be computationally expensive given the high number of output dimensions. Specifically, this approach requires calculating the score function for each input graph $d_y$ times (one for each output dimension), resulting in a total of $d_y \cdot n$ importance scores per input graph.

Additionally, while using equation 2 to find *connected* subgraphs is theoretically possible—by plugging the naive score function from equation 2 into equation 1 to examine all subgraphs and select the one with the highest score—this approach is intractable due to the additional computational complexity of searching for connected subgraphs. In the case of SubgraphX, to manage this additional complexity, we identify important subgraphs for each output dimension separately using the standard implementation and then merge them. That is, instead of directly computing $G^* = \arg\max_{|G^{(i)}| \leq N_{\min}} \left[ \sum_{k=1}^{d_y} f_k(G) \cdot \text{Score}(f_k(\cdot), G, G^{(i)}) \right]$, we identify the important connected subgraphs for each output dimension $k$ individually using $G_k^* = \arg\max_{|G^{(i)}| \leq \hat{N}_{min}} \text{Score}_{\text{Shapley}}(f_k(\cdot), G, G^{(i)})$, and then construct $G^* = \bigcup_{k=1}^{d_y} G_k^*$, where $G_k^*$ is the important subgraph of $k$-th element of GNN output.

While this approach can reduce the computational load (given that $\hat{N}_{\min}$, can be smaller than $N_{\min}$ to achieve the same level of sparsity), it may result in a disconnected subgraph even when a connected subgraph is desired, which could limit the effectiveness and comprehensiveness of the graph explanation.

To address the aforementioned issues, we propose to leverage the linearity property of importance scores with respect to the GNN. This property is common among score functions used for graph explainability, including Shapley, Saliency, and IG (Crabbé & van der Schaar, 2022). Therefore, we propose to rewrite the weighted importance score in equation 2 as follows:

$$\overline{\text{Score}}_{\text{sup}}(f(\cdot), G, G^{(i)}) = \text{Score}(\sum_{k=1}^{d_y} \{f_k(G) \cdot f_k(\cdot)\}, G, G^{(i)}), \tag{3}$$

where $\overline{\text{Score}}_{\text{sup}}$ refers to the scoring function for d-dimensional GNNs output in the supervised settings. The importance score can be computed efficiently through the *auxiliary function* $\psi(G^{(i)}) = \sum_{k=1}^{d_y} f_k(G) \cdot f_k(G^{(i)})$ for all subgraphs $G^{(i)} \in G$. This approach avoids the need to compute the importance scores for each output component individually, thus significantly reducing the computational complexity. Furthermore, it guarantees explanations consisting of connected subgraphs when applied to a score-based method that returns connected subgraphs.

*Remark 1.* The approximation in equation 3 is applicable to most supervised score-based graph explanation methods that compute linear importance scores, including SubgraphX (Yuan et al., 2021). Rather than interpreting individual elements of the argmax output, as normally done in SubgraphX, one can use equation 3 to explain supervised GNNs based on all dimensions of their softmax output. Therefore, we can incorporate this approximation into equation 1 by modifying it as follows:

$$G^* = \arg\max_{|G^{(i)}| \leq N_{\min}} \text{Score}_{\text{Shapley}}(\psi(G^{(i)}), G, G^{(i)}). \tag{4}$$

*Remark 2.* It is noteworthy that equation 3 can also be easily extended to gradient-based models such as Saliency (and IG). In particular, we can calculate the node importance with respect to the GNN output as $\overline{\text{Score}}_{\text{Sal}}(f(\cdot), G, v_i) = \text{Score}_{\text{Sal}}(\sum_{k=1}^{d_y} \{f_k(G) \cdot f_k(\cdot)\}, G, v_i)$, where $\text{Score}_{\text{Sal}}$ is the Saliency value.

## 4.2 Explanation for graph representations

**Notation.** In an unsupervised setting, the trained GNN encoder $f(\cdot)$ maps graphs (or nodes) to a latent space. For graph representation learning, $f(\cdot) : \mathcal{G} \longrightarrow \mathcal{H}$ maps each input graph $G \in \mathcal{G}$ to the latent representation $\mathbf{h}_G = f(G) \in \mathcal{H} \subset \mathbb{R}^{d_h}$, where $1 \ll d_h$. For node embedding, $f(\cdot) : G \longrightarrow \mathcal{H}$ maps each input node $v \in G$ to a latent vector $\mathbf{h}_v = f(v) \in \mathcal{H}$. The goal is to identify important subgraphs that explain the behavior of the GNN and contribute to the latent representation of the graph or node. Ideally, the importance score should reflect the contribution of subgraph $G^{(i)}$ to the representation $\mathbf{h}_G = f(G)$ or $\mathbf{h}_v = f(v)$. However, in this setting, there is no principled way to select a particular output component $f_k(\cdot)$ for some $k \in 1, \ldots, d_h$ to be utilized in the current explainability methods (Figure 1, right).

Equation 3 proposes a framework that offers the benefit of approximating the importance scores of the entire d-dimensional output probability vector, eliminating the need to select a specific output class, which is a common requirement in many existing score-based graph explainability methods. This enables the extension of the same approach to explain graph representations, which is the focus of this paper. However, it is essential to note that while $f_k(G)$ in equation 3 corresponds to the class probability and $\sum_k f_k(G) = 1$ in the supervised setting due to softmax output, the individual dimensions of a latent vector do not necessarily correspond to probabilities.

To address this limitation in our unsupervised setting, we replaced the weighted sum in equation 3 with the *cosine similarity*, which can be seen as a method of normalization and is invariant with respect permutations of the latent dimensions[2].

**Definition 3.2.** Given a trained GNN encoder $f(\cdot)$ that maps graphs to a latent space $\mathcal{H}$, the importance score is defined as:

$$\overline{\text{Score}}_{\text{rep}}(f(\cdot), G, G^{(i)}) \equiv \text{Score}(\psi, G, G^{(i)}), \tag{5}$$

where $\overline{\text{Score}}_{\text{rep}}$ is referring to the score function for graph representation, $\psi : G \longrightarrow \mathbb{R}$ is an auxiliary function defined for all subgraphs $G^{(i)} \in G$ using $\psi(G^{(i)}) = \frac{\sum_{k=1}^{d_h} f_k(G) f_k(G^{(i)})}{|f(G)| \, |f(G^{(i)})|}$, where $d_h$ is the dimensionality of the latent space $\mathcal{H}$. It is important to note that this importance score can also be modified for node representation learning, where the input graph $G$ would be replaced with a single node $v \in G$, and the subgraph $G^{(i)}$ would be a subset of the nodes and edges connected to $v$.

**Extending SubgraphX for graph representation explainability using grXAI (grSubgraphX).** Our framework is applicable to most existing score-based graph explainability methods. In the following, we focus on the generalization of SubgraphX (Yuan et al., 2021). The main reasons for our choice are as follows: 1) SubgraphX is currently SOTA for graph explainability (Yuan et al., 2020; Hajiramezanali et al., 2023), 2) its explanations consist of connected subgraphs, and 3) due to its computational complexity, using it for graph representations necessitates a non-trivial extension.

In SubgraphX, the effectiveness of both the MCTS process and the selection of explanations rely heavily on the accuracy of the chosen scoring function. Therefore, it is crucial to precisely approximate the importance of various subgraphs. To achieve this, we propose to modify SubgraphX using the importance score in equation 5 for explaining graph representations. Let $f(\cdot) : \mathcal{G} \longrightarrow \mathcal{H}$ represent a GNN encoder, where $G = (\mathcal{V}, \mathcal{E}) \in \mathcal{G}$ is a given graph with node set $\mathcal{V} = \{v_1, \ldots, v_N\}$. Let $G^{(s)} = (\mathcal{V}^{(s)}, \mathcal{E}^{(s)})$ be the target subgraph with $N_s$ nodes, where $\mathcal{V}^{(s)} = \{v_1^{(s)}, \ldots, v_{N_s}^{(s)}\}$ is the set of nodes in $G^{(s)}$, and $\mathcal{V}^{(o)} = \mathcal{V} \setminus \mathcal{V}^{(s)} = \{v_1^{(o)}, \ldots, v_{N_o}^{(o)}\}$ is the set of all other nodes not in $\mathcal{V}^{(s)}$, and $N_o = N - N_s$. We define the set of players $P$ as $P = \{G^{(s)}, v_1^{(o)}, \ldots, v_{N_o}^{(o)}\}$, where we consider the entire subgraph $G^{(s)}$ as a single player. In the grXAI version of SubgraphX, named

---

[2]The latent space is not tied to any fixed or predetermined labels on each axis. This means that multiple latent spaces with permuted dimensions can be equivalent to one another. Therefore, the set of transformations that preserve the geometry of the latent space, as determined by cosine similarity, is the set of orthogonal transformations.

*grSubgraphX*, we approximate the Shapley value of the subgraph $G^{(s)}$ as:

$$\varphi_{G^{(s)}}(\psi) = \sum_{R \subseteq P \backslash G^{(s)}} \frac{|R|!(|P| - |R| - 1)!}{|P|!} \left( \psi(R \cup G^{(s)}) - \psi(R) \right),$$

with $\quad \psi : G \longrightarrow \mathbb{R}$ such that for all $G^{(s)} \in G, \quad \psi(G^{(s)}) = S_C \left( f(G), f(G^{(s)}) \right),$ \hfill (6)

where $R$ is a possible coalition set of players, $G^{(s)}$ is a connected graph, and $S_C$ is cosine similarity. Algorithm 1 in Appendix G includes the computation steps for grSubgraphX. The proposed approximation in equation 6 is both fair and accurate, as it considers all possible coalitions and adheres to the four fundamental axioms introduced by Lundberg & Lee (2017): efficiency, symmetry, linearity, and the dummy axiom. These desirable axioms guarantee the validity and impartiality of the explanations for graph representations.

**grSaliency.** Given an input graph $G$ (with $N$ nodes and $M$-dimensional node attributes $\mathbf{X} \in \mathbb{R}^{N \times M}$), and a GNN encoder $f(\cdot) : \mathcal{G} \longrightarrow \mathcal{H}$, we calculate the Saliency score for each node as follows. First, we calculate the derivative $\mathbf{w}_i = \frac{\partial \psi}{\partial v_i} \in \mathbb{R}^M$ by backpropagation, where $\psi$ is cosine similarity define in equation 6. After that, to derive a single Saliency score for each node, we take the magnitude of the average of $\mathbf{w}_i$ across all node features: $\text{Score}_{\text{Sal}}(\psi, G, v_i) = |\text{mean}(\mathbf{w}_i)|$. Finally, we sort the nodes based on their scores and select the nodes with the highest Saliency values as the subgraph. Formally, $G^* = \{v_{\pi(s)}\}_{s=1}^{\mathcal{V}_*}$, where $\pi$ is a permutation of nodes $\{v_i\}_{i=1}^N$ such that $\text{Score}_{\text{Sal}}(\psi, G, v_{\pi(1)}) \geq \text{Score}_{\text{Sal}}(\psi, G, v_{\pi(2)}) \geq \cdots \geq \text{Score}_{\text{Sal}}(\psi, G, v_{\pi(N)})$, and $\mathcal{V}_*$ is the size of desired subgraph.

**grIG.** Similar to grSaliency, we calculate the IG score for each node individually and select nodes with the highest IG score for the explanation. Specifically, we first calculate $\mathbf{w}_i = v_i \int_{\alpha=0}^1 \frac{\partial \psi}{\partial v_i} (\alpha \times v_i) \, d\alpha$. Then, we obtain the IG score for each node as $\text{Score}_{\text{IG}}(\psi, G, v_i) = |\text{mean}(\mathbf{w}_i)|$. Similarly, the identified subgraph is $G^* = \{v_{\pi(s)}\}_{s=1}^{\mathcal{V}_*}$, where $\pi$ is a permutation of nodes $\{v_i\}_{i=1}^N$ such that $\text{Score}_{\text{IG}}(\psi, G, v_{\pi(1)}) \geq \text{Score}_{\text{IG}}(\psi, G, v_{\pi(2)}) \geq \cdots \geq \text{Score}_{\text{IG}}(\psi, G, v_{\pi(N)})$.

## 5 Experiments

To evaluate the effectiveness of the proposed method, we performed experiments on various datasets, graph learning models, and explainability methods. We evaluated our grXAI framework on seven datasets in both unsupervised and supervised settings, covering synthetic, biological, citation network, and text data. The details of the experimental settings are included in Appendix F.

**MUTAG** (Debnath et al., 1991), **BBBP** (Wu et al., 2018), **BACE** (Wu et al., 2018), and **NCI1** (You et al., 2020) are molecular datasets for graph representation learning. Each graph in these datasets represents a molecule, with nodes representing atoms and edges representing bonds. The labels for these datasets correspond to molecular properties and biological activities. **BA-Shapes** (Yuan et al., 2020; 2021) is a synthetic node classification dataset with 4 unique node labels. Each graph includes a base Barabasi-Albert (BA) graph with embedded five-node house-like motifs. The labels for each node are determined by whether it belongs to the base graph or different parts of the motif. The **Graph-Twitter** (Yuan et al., 2020) dataset is a sentiment graph classification dataset with 3 labels. Yuan et al. (2020) convert each tweet sequence into a graph, where each node represents a word and edges are the relationships between words. **Cora** (Sen et al., 2008) is a citation network for node embedding tasks.

### 5.1 Evaluation metrics

As the considered real-world datasets do not provide ground truth for explanations, we follow previous studies (Pope et al., 2019; Xie et al., 2022; Yuan et al., 2020) and adopt Fidelity and Sparsity scores to quantitatively evaluate the explanations.

**Fidelity.** This metric is the main available metric for evaluating post-hoc graph explanations. It assesses whether the input subgraphs identified by the explanation method are important (Yuan et al., 2020). If so, then removing these subgraphs should result in a significant change in the model's outputs. In the supervised setting, Fidelity is calculated as the difference in predicted probability between the original predictions and

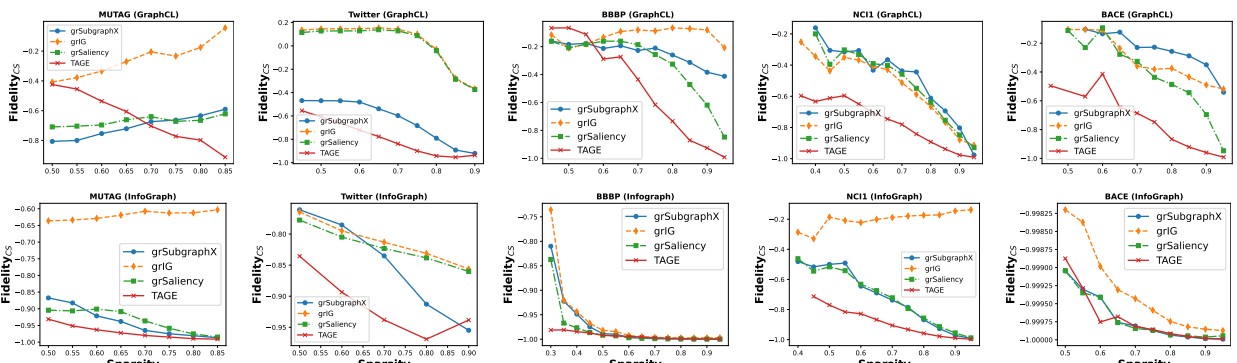

Figure 2: Quantitative results for various explanation techniques, with a preference for higher $\text{Fidelity}_{CS}$ in higher Sparsity levels.

the new predictions after masking out important subgraphs (Pope et al., 2019; Yuan et al., 2021). Formally, given a graph $G$, the softmax output of the predicted class $f_c(\cdot)$, and its explanation as the subgraph $G^*$, we define $G^o = G \setminus G^*$ as the other graph, i.e. the graph of all possible nodes that are not in $G^*$. The Fidelity score can be computed as $\text{Fidelity}(G) = f_c(G) - f_c(G^o)$.

Building on the original definition, we use three types of Fidelity for evaluating explanations for graph representations: (i) **Fidelity**$_{CS}$, which measures the negative *cosine similarity* between the original embedding and the new embedding after masking out the important subgraph. Formally, this is calculated as $-S_c(f(G), f(G^{(o)}))$; (ii) **Fidelity**$_{prob}$, which evaluates the relative importance of the subgraphs on a downstream task. Specifically, it measures the difference between the predicted probability of the original embedding and the new embedding after masking out the important subgraph. The predicted probability is given by a logistic regression model trained on the original embeddings.

**Sparsity.** Effective explanations should be sparse, which means they should capture the most important subgraphs and ignore the irrelevant ones (Yuan et al., 2020). Post-hoc graph explainability methods, including grXAI, can control the size of the identified subgraph through various hyperparameters, and the Sparsity metric measures the proportion of structures identified as important by the explanation method. It is important to note that accounting for Sparsity promotes a fair comparison between different methods. Indeed, larger subgraphs generally improve Fidelity, and therefore explanations with different sizes are not directly comparable. By comparing Fidelity at the same Sparsity, we compare explanations with the same size.

### 5.2 Results and discussion

**Quantitative studies.** We evaluate the quality of graph representations in terms of Fidelity and Sparsity scores. Specifically, we assess the Fidelity of both raw embedding vectors ($\text{Fidelity}_{CS}$) and their downstream tasks ($\text{Fidelity}_{prob}$) by comparing our grXAI-based methods (grSubgraphX, grIG, and grSaliency) with the only available baseline, TAGE (Xie et al., 2022). Please refer to Appendix B.2 for a detailed overview of TAGE. The naive approach (equation 2) is not included due to its significant time cost on real-world datasets. We trained GNN encoders using two different algorithms for self-supervised graph representation learning, InfoGraph (Sun et al., 2019) and GraphCL (You et al., 2020).

The performance of the grXAI framework is demonstrated in Figure 2, which shows better Fidelity (higher $\text{Fidelity}_{CS}$) consistently across all levels of Sparsity. This conclusion is also supported by Table 2 (a higher $\text{Fidelity}_{prob}$ corresponds to better performance). Notably, our gradient-based extensions, particularly grIG, perform exceptionally well on multiple datasets, despite their lower computational complexity compared to grSubgraphX.

The grSubgraphX approach, the only method that explains graph representation based on connected subgraphs, outperforms TAGE on all datasets except Graph-Twitter. This is likely due to the nature of molecules, where localized functional groups significantly affect global molecular properties. Hence, there is a preference for an inductive bias toward explaining based on connected subgraphs. However, sentiment analysis requires a more

Table 2: Fidelity$_{prob}$ scores with controlled Sparsity on a downstream molecular property prediction task based on the graph-level embeddings.

| SSLGraph | Explainer | MUTAG (↑) | Twitter (↑) | BBBP (↑) | NCI1 (↑) | BACE (↑) |
|---|---|---|---|---|---|---|
| GraphCL | grSubgraphX (**ours**) | **0.44± 0.39** | 0.01± 0.17 | **0.68 ± 0.31** | **0.13 ± 0.38** | **0.18 ± 0.53** |
| | grSaliency (**ours**) | 0.39± 0.68 | **0.04± 0.26** | 0.31 ± 0.51 | 0.09 ± 0.41 | 0.07 ± 0.52 |
| | grIG (**ours**) | 0.43± 0.73 | **0.04± 0.26** | 0.14 ± 0.49 | 0.10 ± 0.40 | 0.07 ± 0.48 |
| | TAGE (Xie et al., 2022) | 0.21± 0.33 | 0.01± 0.15 | -0.05 ± 0.24 | 0.07 ± 0.32 | 0.06 ± 0.58 |
| InfoGraph | grSubgraphX (**ours**) | **0.45± 0.60** | 0.04± 0.23 | **0.01± 0.03** | **0.15± 0.29** | **0.25± 0.30** |
| | grSaliency (**ours**) | 0.38± 0.55 | 0.11± 0.33 | 0.00± 0.02 | 0.13± 0.33 | 0.14± 0.11 |
| | grIG (**ours**) | **0.45± 0.59** | **0.12± 0.34** | 0.00± 0.03 | 0.14± 0.34 | 0.10± 0.81 |
| | TAGE (Xie et al., 2022) | 0.36± 0.52 | 0.04± 0.21 | **0.01± 0.21** | 0.10± 0.30 | 0.06± 0.89 |

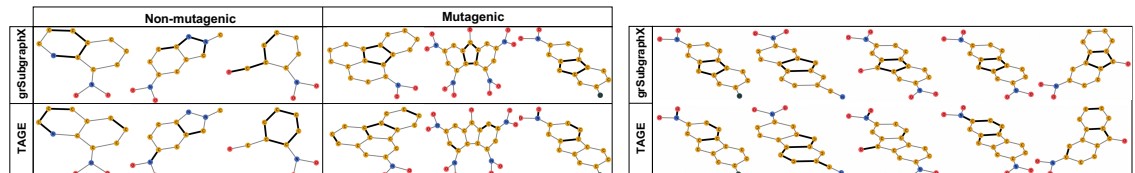

Figure 3: **(Left)** The explainability of graph representations for six molecules from the MUTAG dataset. Carbon, Oxygen, and Nitrogen atoms are highlighted in yellow, red, and blue, respectively. The top row displays the grSubgraphX explanation, which is a connected subgraph that is more easily interpretable for molecular graphs. In particular, the grSubgraphX explanation often identifies a carbon ring for mutagenic compounds, which is consistent with the prior knowledge of the underlying chemistry. **(Right)** The explainability of graph representations for five molecules with minor structural changes from the MUTAG dataset. The grSubgraphX method generates highly stable explanations, while TAGE is not as stable, i.e., small changes to the molecular graph produce substantially different explanations.

flexible approach to handling disconnected subgraphs. Our gradient-based techniques, grIG and grSaliency, outperform grSubgraphX in such cases since they do not have any constraints on providing connected subgraphs. This flexibility allows them to perform better in scenarios where disconnected subgraphs play a vital role in the task.

In addition to Fidelity$_{CS}$, we also compare grSubgraphX to TAGE based on the Infidelity$_{CS}$ metric, where Infidelity$_{CS} = -S_c\left(f(G), f(G^{(*)})\right)$. While Fidelity$_{CS}$ measures how much the representation changes when *important* subgraphs are masked, Infidelity$_{CS}$ directly measures how much the *important* subgraphs contribute to the representation. In contrast to Fidelity$_{CS}$, lower Infidelity$_{CS}$ is desired, meaning that the representation of the identified important subgraph is close to the original graph representation. Figure A1 shows the results on this metric. As shown, grXAI outperforms the baseline in terms of Infidelity$_{CS}$, particularly in regions of higher sparsity. In most cases, the performance is significantly better across a range of sparsity levels. This is mostly because according to equations 1 and 6, our model's optimization to find the important subgraph is based on minimizing the Infidelity metric.

**Qualitative studies.** To determine the effectiveness of explaining by connected subgraphs in real-world molecule datasets, we conducted further investigations using the MUTAG dataset, for which a deeper understanding of the underlying mechanism connecting structural features to the property is available (Yuan et al., 2021). Specifically, it is known that carbon rings tend to be mutagenic (Debnath et al., 1991). We study whether the explanations provided by different methods can identify the carbon rings characterizing the positive class. Our results, presented in Figure 3 (left), demonstrate that 1) our grSubgraphX successfully and precisely identifies the carbon rings as important subgraphs for the mutagenic class; and 2) grSubgraphX provides more localized and interpretable explanations for both classes. This is a critical factor in understanding molecular properties and aiding scientific decision-making, making grSubgraphX a valuable tool for the field.

**Stability study.** There have been arguments that post-hoc (graph) explainability methods may lack stability (Adebayo et al., 2018), as even negligibly slight changes to an instance can lead to significantly different explanations (Hajiramezanali et al., 2023). To test the stability of the grSubgraphX method, we compare it with TAGE using the real-world MUTAG dataset. Results in Figure 3 (right) and Figure A5 indicate that grSubgraphX is able to provide consistent explanations for molecules with minor structural changes, while TAGE produces radically different results in those cases, thus highlighting the stability of our framework. The stability of grSubgraphX is due to its ability to evaluate the importance of subgraphs as a set of interconnected nodes instead of considering the importance of individual edges separately, as is done in TAGE.

**Efficiency study.** Our framework significantly decreases the computational complexity associated with graph representation explainability from $O(\text{graphXAI}) \times d_h$ to $O(\text{graphXAI})$, where $O(\text{graphXAI})$ is the computational complexity of the original graph explainability method, and $d_h$ denotes the number of hidden dimensions (e.g., 512 in the case of InfoGraph). To assess the efficiency of our proposed grSubgraphX, we follow the methodology employed in (Yuan et al., 2021; Zhang et al., 2022) and calculate the average time taken to generate explanations for graphs from the BBBP dataset, with each graph containing an average of 24.04 nodes.

Table 3 shows that the naive calculation of SubgraphX (equation 2) to explain the embedding of InfoGraph requires approximately 11 hours per graph. In contrast, our proposed grSubgraphX method accomplishes the same task in less than 2 minutes per graph. Consequently, utilizing the naive approach would take a minimum of 916 days to explain a small molecule dataset comprising 2k samples; however, grXAI archives it within 2 days.

Table 3: Average running time for explaining graph representation of InfoGraph for BBBP.

| **Method** | Naive SubgraphX | grSubgraphX **(ours)** |
|---|---|---|
| TIME | > 11 hours | 106.32 sec |

**Explanation for node embedding.** Although we have focused on graph-level learning tasks, our grXAI framework also applies to node embedding tasks. Figure 4 shows a GNN trained by the GRACE (Zhu et al., 2020) algorithm on the Cora dataset (Sen et al., 2008), with explanations given by grSubgraphX (top) and TAGE (bottom). Qualitatively, we observe that grSubgraphX identifies neighbor nodes that are densely connected and have the same class as the target node (i.e., the node we are explaining). In contrast, TAGE explanations are more disconnected, scattered, and difficult to interpret. Quantitatively, grSubgraphX achieves better Fidelity$_{CS}$ scores across varying Sparsity values (Figure A2), demonstrating that our framework also improves the performance in node embedding tasks.

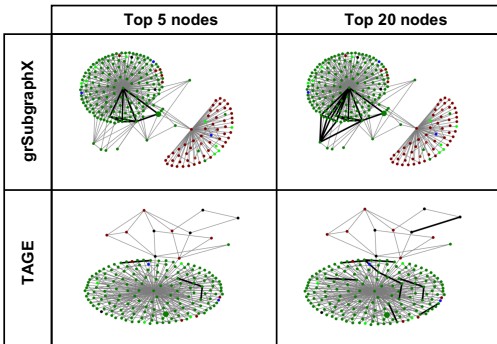

Figure 4: Explanations for a specific node in the Cora dataset. The plots show a part of the Cora graph, highlighting the target node and its 2-hops neighboring nodes. The target node is shown as a larger green circle, with different colors denoting node labels. The edges connecting the most important nodes are bold.

### 5.2.1 grXAI in supervised settings

Although grXAI has been primarily designed to improve performance and efficiency in explaining unsupervised GNNs, a natural question is whether it maintains the same performance in supervised settings. In the following, we show that not only grXAI versions of the explainability methods achieve identical performance to their original counterparts with little additional computational overhead, but they also surpass them in cases with high-uncertainty predictions. Specifically, we compare our grXAI modifications (grIG, grSaliency, and grSubgraphX) to their original counterparts on a variety of datasets and tasks, including BBBP, MUTAG, and Twitter datasets for graph classification and BA-shape for node classification. As expected, our grXAI results achieve equivalent performance in this supervised setting (Figure A3, Appendix). This is mainly because, in most cases, the GNNs generate highly confident predictions, i.e., $f_c(G_i) \approx 1$ (or $f_c(v_i) \approx 1$ in

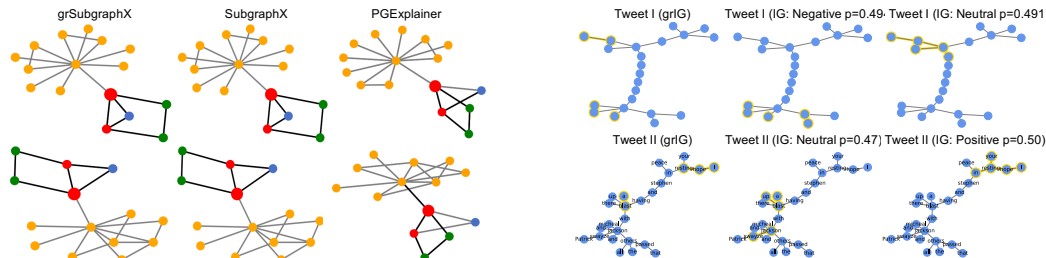

Figure 5: grXAI in supervised settings. **(Left)** Explanation results on the BA-Shape dataset. The target node is shown in a larger size. Different colors denote node labels. Each graph contains a base graph (yellow) and a house-like five-node motif. grSubgraphX (similar to SubgraphX) precisely identifies the motifs as the explanations. **(Right)** Explanation of a GCN classifier for the Graph-Twitter dataset. The blue nodes represent words. Important subgraphs (edges and nodes) are highlighted in yellow. In the first tweet, we removed the words as the tweet was an opinion about a specific person.

node classification). Node classification results on BA-Shape dataset are shown in Figure 5 (left), with the important substructures highlighted in bold. We observe that the grSubgraphX identifies motifs that are consistent with the ground truth.

To investigate *when* grXAI outperforms the original counterparts in supervised settings, we examined specific examples from the Twitter dataset with high uncertainty predictions, i.e., $f_c(G_i) \approx 0.5$. Figure 5 (right) shows IG and grIG explanations for two such examples. grIG simultaneously explains both predictions with two subgraphs that correspond to both labels with a similar probability. Instead, IG explainability changes depending on whether the true class or the predicted class is used. This illustrates that traditional methods based on a single output may not always accurately and reliably explain the GNN, particularly in cases of *high uncertainty*, whereas our d-dimensional vector-based approach can provide explanations that directly reflect a soft distribution over multiple classes.

# 6   Discussion

In this paper, we introduced grXAI, a novel framework to extend score-based graph explainability methods to unsupervised GNN settings. We analyzed our framework and demonstrated its compatibility with many score-based methods, thus highlighting its flexibility. We validated grXAI applied to different self-supervised graph representation learning, self-supervised node embedding, and supervised models across several datasets, both qualitatively and quantitatively. Overall, grXAI leads to accurate and stable explanations which can be computed efficiently and outperform previously-proposed method. We also investigated the impact of different augmentation strategies on the explainability of graph representations (Appendix D). This new perspective can help domain experts to identify which augmentation technique is more closely related to the main molecular property of interest, leading to semantically meaningful design choices, and ultimately enhancing the effectiveness of self-supervised graph representation learning methods.

While our framework has demonstrated success in explaining graph representations in an efficient way, it is important to acknowledge its limitations. The grXAI framework identifies subgraphs that increase the similarity to the original graph in the representation space from any direction. However, in practical applications, some of these directions may not be meaningful or equally important. Instead, the goal may be to identify subgraphs that increase the similarity to a target graph with desired properties, while simultaneously separating them from a set of graphs with undesirable properties in the representation space, such as non-mutagenic molecules. This is similar to how human perception operates (Lin et al., 2023). We believe grXAI opens the door to several exciting avenues for future research, such as developing example-based explainability of graph representation by examining similarity to other graphs' representations (Lin et al., 2023), and extending grXAI to the structure-aware scoring functions, for instance, by adapting GStarX (Zhang et al., 2022).

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

# A   Details of datasets

We evaluated our grXAI framework on seven datasets in both unsupervised and supervised settings, covering synthetic, biological, citation network, and text data. The datasets are summarized as follows:

**Graph-Twitter** (Yuan et al., 2020) dataset is a sentiment graph classification dataset with 3 labels. Yuan et al. (2020) convert each tweet sequence into a graph, where each node represents a word and edges represent the relationships between words. Biaffine parser (Gardner et al., 2018) has been used to extract word dependencies, and a pre-trained 12-layer BERT (Devlin et al., 2018) model has been used to extract a 768-dimensional feature vector for each word.

**BA-Shape** (Yuan et al., 2020) is a synthetic node classification dataset with 4 labels. Each graph contains a base graph (300 nodes) and a house-like five-node motif. The base graph is obtained by the Barabási-Albert (BA) model, which can generate random scale-free networks with a preferential attachment mechanism. The motif is attached to the base graph while random edges are added. Each node is labeled based on whether it belongs to the base graph or to different spatial locations of the motif.

**Molecule dataset.** Molecular datasets such as MUTAG, BBBP, BACE, and NCI1 are widely used in explanation and graph representation learning tasks (You et al., 2020; Yuan et al., 2021; Sun, 2022). Each graph in such datasets corresponds to a molecule where nodes represent atoms and edges are the chemical bonds. The labels of molecular graphs are generally determined by the chemical functionalities or properties of the molecules. In particular, in the MUTAG dataset, molecular graphs are labeled based on their mutagenic effects on a bacterium. It is known that carbon rings and NO2 chemical groups may lead to mutagenic effects (Yuan et al., 2021; Zhang et al., 2022).

**Cora** (Sen et al., 2008) is a citation network for node embedding tasks with seven different classes.

# B   Additional preliminaries

## B.1   Graph self-supervised learning

We applied our explainer to two self-supervised graph-level representation learning models:

**GraphCL** (You et al., 2020). GraphCL is a contrastive learning framework for learning unsupervised representations of graph data. GraphCl proposes several graph data augmentations as well as a novel graph contrastive learning framework for GNN pre-training.

**InfoGraph** (Sun et al., 2019). Infograph maximizes the mutual information between the graph-level representation and the representations of different components with different scales (e.g., nodes, edges, triangles). By doing so, the graph-level representations encode aspects of the data that are shared across different scales of substructures.

We also used **GRACE** (Zhu et al., 2020) for pre-training GNN encoders in node-level self-supervised settings. GRACE is an unsupervised graph representation learning framework that leverages a contrastive objective at the node level. Specifically, it generates two graph views by corruption and learns node representations by maximizing the agreement of node representations in these two views.

## B.2   Task-agnostic GNN explainer

TAGE, introduced by Xie et al. (2022), decomposes a supervised GNN into an encoder model and a downstream model, designing separate explainers for each component. The embedding explainer part, which is the only available baseline for explaining graph representations, is trained using a self-supervised training framework.

TAGE aims to maximize the mutual information (MI) between two embeddings: one from the input graph $G$ and one from the corresponding important subgraph $G^*$ induced by the explainer. It introduces a masking vector $\mathbf{p}$ to indicate specific dimensions of the embeddings to maximize the MI. During training, TAGE samples the masking vector $\mathbf{p}$ from a multivariate Laplace distribution to exploit sparse gradients, ensuring that only a few dimensions are of high importance (Xie et al., 2022). This assumes that embeddings from

well-trained GNN encoders are informative with low dimension redundancy. Formally, the learning objective based on the restricted MI is:

$$\max_\theta E_{\mathbf{p}} \left[ \mathbf{MI} \left( \mathbf{p} \otimes f(G), \mathbf{p} \otimes f(\tau_\theta(\mathbf{p}, G)) \right) \right],$$

where $\mathbf{MI}(\cdot, \cdot)$ computes the mutual information between two random vectors, $\mathbf{p}$ denotes the random masking vector sampled from a certain distribution, $\tau_\theta(\mathbf{p}, G)$ identifies the subgraph of high importance, and $\otimes$ denotes the element-wise multiplication, which applies masking to the embeddings $f(\cdot)$. Intuitively, given an input graph $G$ and the desired embedding dimensions to be explained, the explainer $\tau_\theta$ predicts the subgraph whose embedding shares the maximum mutual information with the original embedding on the desired dimensions (Xie et al., 2022).

To efficiently compute the mutual information, TAGE uses a contrastive loss such as InfoNCE. Additionally, to restrict the size of the subgraphs provided by the explainer, TAGE includes a size regularization term (Xie et al., 2022). TAGE employs a multilayer perceptron (MLP) as the base architecture of the explainer $\tau_\theta$, which predicts the importance score for each edge, leading to disconnected subgraphs.

## C  Additional experiments

**Explanation for graph representations.**  Figure A1 compares grSubgraphX and TAGE based on the Infidelity$_{CS}$ metric, where Infidelity$_{CS} = -S_c\left(f(G), f(G^{(*)})\right)$. Infidelity$_{CS}$ directly measures how much the *important* subgraphs contribute to the representation; therefore, lower Infidelity$_{CS}$ means the representation of the identified important subgraph is close to the original graph representation. As shown, grXAI outperforms TAGE in terms of Infidelity$_{CS}$ in most cases. This is mostly because according to equations 1 and 6, grSubgraphX optimization is based on the Infidelity metric.

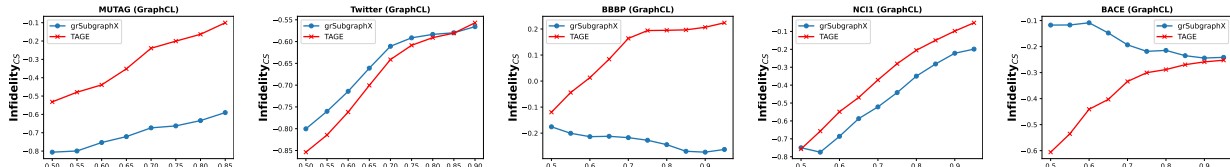

Figure A1: Quantitative results for various explanation techniques, with a preference for lower Infidelity$_{CS}$ in higher Sparsity levels.

**Explanation for node embedding.**  As we stated in the main paper, grSubgraphX has a better explanation performance in terms of both Fidelity$_{CS}$ and visualization. For evaluating this, we trained a GNN using GRACE (Zhu et al., 2020) on the citation network Cora. Then, we compare the performance of grSubgraphX to the baseline (TAGE) using test nodes that have at least 50 nodes in their 2-hop subnetworks. Figure A2 also shows that grSubgraphX outperforms TAGE in terms of Fidelity$_{CS}$ for different Sparsity levels. Please note that due to the higher number of classes in this dataset and the fact that the node embedding has been calculated as the output of a two-layer GNN, the change in Fidelity$_{CS}$ is lower here compared to other experiments.

**grXAI in supervised settings.**  We compare our grXAI-based methods, which employ d-dimensional softmax outputs (grIG, grSaliency, and grSubgraphX), with their original counterparts that rely on explaining the predicted class using the BBBP dataset (as shown in Figure A3). Our proposed grXAI framework has identical performance to its original counterparts but is more efficient, especially for high-confidence predictions. This is because supervised GNNs are often overconfident in their predicted class, with $f_c(G_i) \approx 1$, and therefore taking into account all the outputs does not significantly affect the performance.

**Effect of the wrapper function.**  In unsupervised settings, the individual dimensions of a latent vector do not necessarily correspond to probability vectors. To address this, we replaced the weighted sum in Equation 3 with a *cosine similarity*, which can be seen as a method of normalization that is invariant with respect to latent symmetries. We also experimentally evaluate this selection compared to the dot-product.

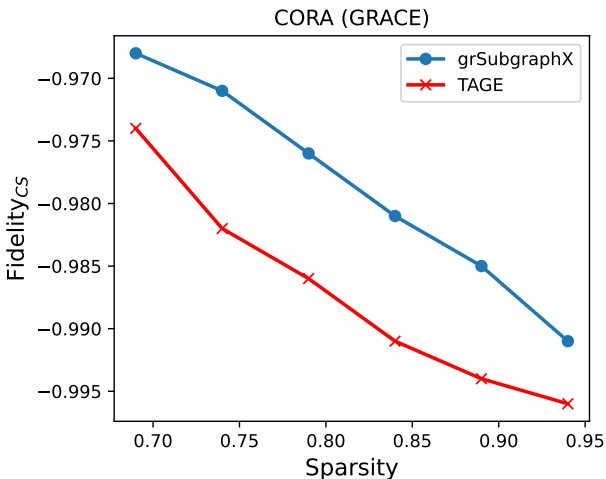

Figure A2: Quantitative performance comparisons between TAGE and grSubgraphX on Cora dataset, with a preference for higher Fidelity$_{CS}$ in higher Sparsity levels.

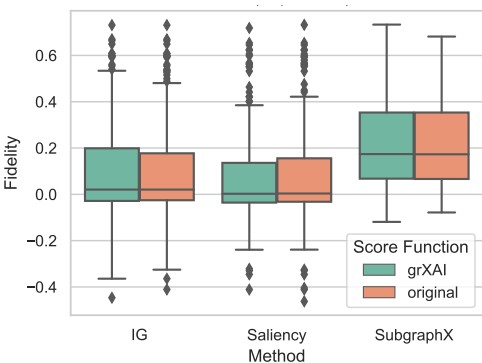

Figure A3: Fidelity comparisons of our proposed grXAI-based methods with their original implementation on supervised task for BBBP dataset.

Our results show that the wrapper function has little effect on the performance of the grSubgraphX. In some datasets, cosine similarity yields slightly better average results, as seen in Figure A4. This might be due to the fact that the latent vector typically corresponds to the activation functions of the neurons in the GNN (e.g. ReLU or sigmoid), which reduce the weight of inactivated components on the score function (Crabbé & van der Schaar, 2022).

**Stability study.** As we stated in the main paper, there have been arguments that post-hoc (graph) explainability methods may lack stability (Slack et al., 2021; Adebayo et al., 2018), as even slight changes to an instance can lead to significantly different explanations. In addition to Figure 3 (right) in the main paper, our results in Figure A5 show that grSubgraphX is able to provide very stable explanations for molecules with minor structural differences, while TAGE generates completely different explanations.

## D Augmentation design

In self-supervised settings, understanding the augmentation strategies can inform the design of novel learning techniques that are not solely based on empirical approaches. For instance, the GraphCL (You et al., 2020) framework introduced various types of graph augmentations, each of which incorporates specific prior knowledge about graph data. Results such as those shown in Figure A6 and Table A1, demonstrate that

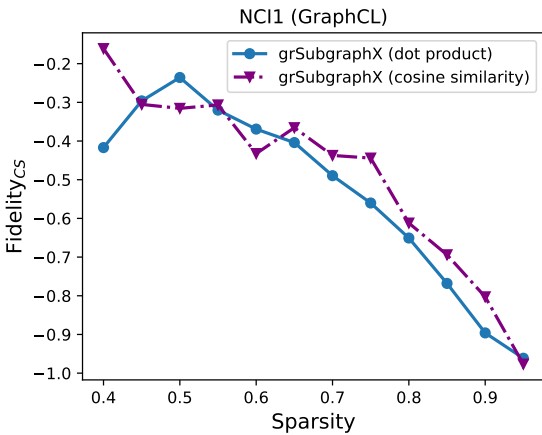

Figure A4: Comparison of wrapper function performance on the NCI1 dataset for grSubgraphX explainability.

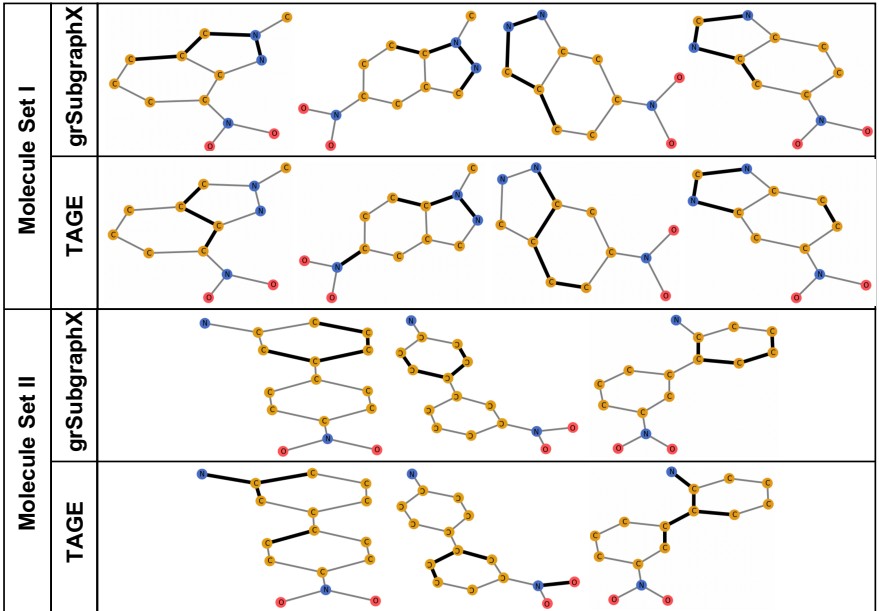

Figure A5: The explainability of graph representations for two sets of molecules with minor structural changes from the MUTAG dataset. The grSubgraphX generates highly stable explanations, while TAGE is not stable, i.e. small changes to the sample produce substantially different explanations.

different augmentations used in GraphCL training correspond to different explanation subgraphs, and have different Fidelity, meaning that perturbing them will affect the representation learning differently. This insight can aid in understanding the effectiveness of self-supervised learning on graph data and provide a basis for comparing different pre-training methods beyond their predicted performance.

# E   Computational complexity

We further compare the computational complexity of our grXAI-based methods to their traditional counterparts on various supervised node and graph classification datasets. As seen in Table A2, our proposed methods have close computational complexity to their traditional counterparts and are significantly faster than averaging over all output components. We should also point out that providing a connected subgraph

Table A1: Comparison of the Fidelity$_{CS}$ on the explainability of GraphCL using different data augmentations on MUTAG dataset.

| Augmentation | grSubgraphX ($\uparrow$) | grIG ($\uparrow$) | grSaliency ($\uparrow$) |
|---|---|---|---|
| dropN | -0.55$\pm$ 0.83 | -0.31$\pm$ 0.14 | -0.69$\pm$ 0.13 |
| maskN | -0.42$\pm$ 0.90 | -0.20$\pm$ 0.19 | -0.46$\pm$ 0.21 |
| permE | -0.52$\pm$ 0.85 | -0.21$\pm$ 0.13 | -0.70$\pm$ 0.16 |
| subgraph | -0.50$\pm$ 0.86 | -0.16$\pm$ 0.09 | -0.62$\pm$ 0.15 |
| random2 | -0.61$\pm$ 0.15 | -0.22$\pm$ 0.18 | -0.65$\pm$ 0.14 |

Figure A6: The explanation for graph representations of two different MUTAG molecules learned by GraphCL. Each column corresponds to a different augmentation technique to train GraphCL. Using different augmentation pushes the model to learn the representation based on different substructures of input molecules, and some of them might be more aligned with prior knowledge.

to explain GNN encoders using grSubgraphX comes at the cost of higher computational complexity. In cases where computational complexity is a bottleneck, we suggest applying our gradient-based extensions, i.e. grIG and grSaliency.

Table A2: Comparison of the performance of our proposed methods to their traditional counterparts in terms of average run time (sec) for exampling one input graph.

| Explainer | BBBP | Twitter | BA-Shape |
|---|---|---|---|
| IG | 0.25 $\pm$ 0.00 | 0.262 $\pm$ 0.00 | – |
| IG$_{avg}$ | 0.51 $\pm$ 0.01 | 0.786 $\pm$ 0.03 | – |
| grIG | 0.27 $\pm$ 0.00 | 0.264 $\pm$ 0.00 | – |
| Saliency | 0.0052 $\pm$ 0.00 | 0.0052 $\pm$ 0.00 | – |
| Saliency$_{avg}$ | 0.0103 $\pm$ 0.00 | 0.0156 $\pm$ 0.00 | – |
| grSaliency | 0.0057 $\pm$ 0.00 | 0.0052 $\pm$ 0.00 | – |
| SubgraphX | 82.41 $\pm$ 173.64 | 80.50$\pm$ 21.09 | 0.26 $\pm$ 0.17 |
| SubgraphX$_{avg}$ | 164.82 $\pm$ 348.28 | 241.5$\pm$ 65.39 | 1.05 $\pm$ 0.68 |
| grSubgraphX | 106.32 $\pm$ 224.09 | 84.53 $\pm$ 23.76 | 0.41 $\pm$ 0.24 |

## F  Experimental settings

We used PyTorch (Paszke et al., 2019) to develop our grXAIe framework and conducted experiments on an Nvidia A100 with 80 GB of memory. For supervised experiments, we used GNN checkpoints from DIG (Liu et al., 2021) as part of Yuan et al. (2020) and did not train any new models. We also utilized the DIG package's

implementation of GraphCL, InfoGraph, and GRACE for pre-training GNNs in self-supervised settings (Liu et al., 2021), specifically using GIN (Xu et al., 2019) for GraphCL and InfoGraph, and GCN for GRACE. We followed the hyperparameters outlined in the main papers for these methods, using 3-layer GINs with an embedding dimension of 32 for GraphCL, 4-layer GINs with an embedding dimension of 512 for InfoGraph, and 2-layer GCNs with a node embedding dimension of 128 for GRACE. For grSubgraphX, we modified the original implementation of SubgraphX (Yuan et al., 2021) and followed the same hyperparameters of the original paper. We have implemented grIG and grSaliency based on their original implementation in the Captum (Kokhlikyan et al., 2020) package.

## G   grSubgraphX algorithm

Algorithm 1 shows how we calculate the Shapley score in our grXAI framework.

---

**Algorithm 1** The algorithm of grSubgraphX to calculate Shapley score.

---

**Input:** $L$-layer GNN model $f(\cdot)$, input graph $G$ with nodes $\mathcal{V} = \{v_1, \ldots, v_N\}$, subgraph $G^{(s)} = (\mathcal{V}^{(s)}, \mathcal{E}^{(s)})$ with nodes $\mathcal{V}^{(s)} = \{v_1^{(s)}, \ldots, v_{N_s}^{(s)}\}$, Monte-Carlo sampling steps $T$.

**Initialization:** Obtain the $L$-hop neighboring nodes of $G^{(s)}$, denoted as $\mathcal{V}^{(L_s)} = \{v_1^{(L_s)}, \ldots, v_{N_{L_s}}^{(L_s)}\}$. Then the set of players is $P' = \{G^{(s)}, v_1^{(L_s)}, \ldots, v_{N_{L_s}}^{(L_s)}\}$.

**for** $t = 1$ **to** $T$ **do**

    Sampling a coalition set $R_t$ from $P' \setminus G^{(s)}$.

    Set nodes from $V \setminus (R_t \cup G^{(s)})$ with zero features and feed to the GNNs $f(\cdot)$ to obtain $\psi(R_t \cup G^{(s)}) = \frac{\sum_j f_j(G) f_j(R_t \cup G^{(s)})}{\|f(G)\| \|f(R_t \cup G^{(s)})\|}$.

    Set nodes from $V \setminus R_t$ with zero features and feed to the GNNs $f(\cdot)$ to obtain $\psi(R_t) = \frac{\sum_j f_j(G) f_j(R_t)}{\|f(G)\| \|f(R_t)\|}$.

    Then $\varphi_t = \psi(R_t \cup G^{(s)}) - \psi(R_t)$.

**end for**

**Return:** $\text{Score}(f(\cdot), G, G^{(s)}) = \frac{1}{T} \sum_{t=1}^{T} \varphi_t$.

---

