# OpenReview forum: "Score-based Explainability for Graph Representations"
_TMLR — Accepted by TMLR_

### Review · Reviewer_fYbL · 2024-07-23

**Summary Of Contributions:**

This paper proposes a post-hoc explanation method to find the most important connected input subgraph to the hidden embedding vector. The method mainly extends SubgraphX to unsupervised settings and show good results in quantitative and qualitative evaluations.

**Audience:**

Yes

**Claims And Evidence:**

Yes

**Requested Changes:**

1.	It is better to explain why the function $Shapley(\cdot, \cdot, \cdot)$ in Equation (1) has three inputs, and how these inputs are related to the game-theoretic definition of Shapley values (e.g., which input corresponds to the value function, and which input corresponds to the set the players, etc.).

2.	Could you explain why a lower $Fidelity_{CS}$ score is considered better? First, it sounds quite counter-intuitive to claim that lower fidelity is better. Second, according to the definition of $Fidelity_{CS}$, it is the “difference between the cosine similarity of the original embedding (with the original embedding itself) and (the cosine similarity of) the new embedding after masking out the important subgraph (with the original embedding)”. Therefore, a lower $Fidelity_{CS}$ score implies that the identified subgraph is not important to the embedding vector, which means an undesirable explanation result. This conflicts with the claim in the paper.

3.	In addition to the fidelity $Fidelity(G) = f_c(G) – f_c(G^o)$, could you also run experiments on the *infidelity*, which is analogously defined as $Infidelity(G) = f_c(G) – f_c(G^*)$? In the above definitions, a higher fidelity does not imply a lower infidelity, since the function $f_c$ is not linear. The fidelity metric measures how much the prediction confidence drops when “important” subgraphs are masked, while the infidelity defined above directly measures how much the “important” subgraph contributes to the prediction confidence. In this sense, the latter should not be ignored.

**Strengths And Weaknesses:**

Strengths:

1.	The method extends score-based explanations to unsupervised settings, where the output of the model is a d-dimensional vector without clear semantic meanings on each dimension.

2.	The method produces connected subgraph, which is better for human interpretation.

3.	The presentation is clear and easy to follow.


Weaknesses:

1.	In Equation (5), using cosine similarity as a measure of similarity between two embedding vectors can be problematic. A high cosine similarity can imply the two embedding vectors are close to each other, but a low (especially negative) cosine similarity does not necessarily mean the two embedding vectors are dissimilar. For example, if the two embedding vectors is only different in their signs (i.e., they are anti-parallel), the cosine similarity is -1, which is the lowest value one can achieve with cosine similarity. However, there is no reason to assert that the two embedding vectors are dissimilar, as one only need to change the sign of weights in subsequent layers to yield the same output.

2.	In the paragraph following Equation (6), how can one ensure that $R \cup G^{(s)}$ is always a connected graph. As shown in Equation (6), $R$ is any set of nodes in $P\setminus G^{(s)}$, so it is likely that $R \cup G^{(s)}$ forms a disconnected graph.

---

> ### Author Response · Authors · 2024-08-14
>
> Thank you very much for taking time to review our paper and let us know your valuable comments. Please find below our responses to your comments and questions.
>
> **Re W1:** We would like to clarify that this is actually the reason why we leveraged cosine similarity. Cosine similarity is a good measure for evaluating latent embeddings because it focuses on the directionality of the vectors, making it robust to scaling and effective for capturing high dimensional latent similarity. For these reasons, cosine similarity plays a key role in representation learning approaches (You et al., 2020). For example, contrastive learning (typically through the InfoNCE loss) is used to maximize the cosine similarity between positive pairs (augmentations of the same graph) and minimize the cosine similarity between negative pairs (augmentations of different graphs).
>
> **Re W2:** Thank you for pointing this out. $R\cup G^{(s)}$ is not necessarily a connected subgraph; rather, it is a combination of a coalition set $R$ and a connected subgraph $G^{(s)}$. We will make sure to rewrite this in the revised version.
>
> **Re Q1:** Sorry for the confusion with the notation. Shapely() in Equation (1) does not refer to the game-theoretic definition of Shapley value. Rather, we used this notation to refer to the score function based on Shapley. We will change the notation to $Score_{Shapely}$ in the revised version to address the issue. Regarding your question about the connection with the game-theoretic definition of Shapley values in graph representation, we use $\psi$ as the game gain and different graph structures as players, as we stated in Equation (6) in the paper.
>
>
> **Re Q2:** Thank you for the comment. Fidelity$_{CS}$ is actually the cosine similarity between the original embedding and the new embedding after masking out the important subgraph. Specifically, we calculate $S_c(f(G), f(G^{(o)}))$. If the identified subgraph is not important, then the new embedding after masking out the important subgraph will not change significantly and therefore $S_c \rightarrow 1$. If the subgraph is important, then the new embedding will be far from the original one, and $S_c \rightarrow -1$. We will improve the definition on page 7, also removing "**the difference between**", to make it clearer.
>
>
> **Re Q3:** Thank you for the suggestion. The main reason we did not use this metric for evaluation is that InFidelity actually benefits our proposed method, making it potentially unfair to use it as an evaluation metric. As seen in equations (1) and (6), our model’s optimization to find the important subgraph is based on minimizing the InFidelity. We are open to running such experiments if the reviewer still advises us to do so.

---

> > ### Comment · Reviewer_fYbL · 2024-08-23
> >
> > Thank you for your response and clarification. Here are some further comments after I read the response:
> >
> > 1. It is better to revise the definition of Fidelity so that a higher Fidelity score implies a better explanation result. This prevents unnecessary confusion (e.g., the readers may not see the claim "lower fidelity is better").
> >
> > 2. I still recommend the authors run experiments using the *infidelity*  metric I mentioned to make the evaluation more comprehensive and convincing. If you think this metric benefits the proposed method and yields biased results, you could add a discussion about this issue and why it happens.
> >
> > Lastly, I will take the response into consideration when making my final recommendation.

---

> ### Author Response · Authors · 2024-08-26
>
> We appreciate your thoughtful feedback and valuable suggestions. We agree that revising the definition of Fidelity_CS to ensure that a higher Fidelity score implies a better explanation result will help prevent confusion. Therefore, we have modified the definition of Fidelity_{CS} to $- S_c(f(G), f(G^{(o)}))$ (i.e., the opposite of the previous definition). With this change, a higher Fidelity$_{CS}$ score will indeed indicate a better explanation result. We have updated Figure 1 accordingly to reflect this adjustment.
>
> In addition, we have incorporated the **infidelity** metric into our evaluation. We have compared the performance of grXAI with TAGE according to this metric in Figure A1 in the revised version. We have also included a discussion on why grXAI tends to perform better on this metric in the last paragraph of the quantitative studies section on page 6.
>
> Thank you for your comments. We believe these updates will enhance the comprehensiveness and clarity of our paper.

---

### Review · Reviewer_7f3w · 2024-07-28

**Summary Of Contributions:**

The paper studies the problem of explaining unsupervised Graph Neural Networks (GNNs), which produce d-dimensional representation vectors without clear semantic meaning. The authors propose a novel framework called grXAI, which generalizes existing score-based graph explainers to identify the subgraph most responsible for constructing the latent representation of the input graph. This framework can be implemented as a wrapper around existing methods, making the explanations more human-intelligible by focusing on connected subgraphs. Extensive qualitative and quantitative experiments demonstrate grXAI’s ability to effectively explain learned graph representations across various unsupervised tasks and learning algorithms.

**Audience:**

Yes

**Broader Impact Concerns:**

No concerns.

**Claims And Evidence:**

Yes

**Requested Changes:**

See the weaknesses above.

**Strengths And Weaknesses:**

**Strengths**

1. Novel Framework for Unsupervised GNNs.

The idea of grXAI is quite novel in the field of graph explainability, specifically tailored for unsupervised GNNs. Existing methods are primarily designed for supervised settings, leaving a gap in explainability for unsupervised models. grXAI fills this gap by providing a robust method to interpret d-dimensional representations.

2. Human-Intelligible Explanations.

By focusing on connected subgraphs, grXAI provides explanations that are more intuitive and understandable to humans. Human-friendly explanations are crucial for practical applications, as they help users comprehend the model's behavior and make informed decisions based on its outputs.

3. Compatibility with Existing Methods.

The framework can be easily integrated with current score-based graph explainers, enhancing their applicability to unsupervised settings without significant modifications. This compatibility ensures that the framework can be widely adopted and used with minimal additional effort from researchers and practitioners.

**Weaknesses**

1. Computational Complexity.

Despite improvements, the method still involves significant computational resources, particularly for complex models and large datasets. High computational demands can limit the framework's accessibility and scalability, posing challenges for researchers with limited resources.

2. Specificity to Predefined Node and Edge Types.

The framework requires predefined node and edge types, restricting its flexibility and generalizability to new types of graphs without retraining. This limitation reduces the model's adaptability to different domains or novel graph types, hindering its broader applicability.

3. Lack of Real-world Case Study.

The paper lacks a detailed case study to showcase the practical effectiveness and real-world impact of the proposed method. A case study would provide concrete evidence of the method's utility in solving real-world problems, enhancing the paper's impact and credibility.

4. Focus on Synthetic Data.

The reliance on synthetic and controlled datasets may not fully capture the complexity and variability of real-world misuse scenarios, potentially limiting the generalizability of the findings. Real-world data often present unique challenges that synthetic data cannot replicate, making it essential to validate findings in more practical settings.

5. Small Dataset Sizes.

Most of the datasets used in the experiments are relatively small, which may not adequately represent the performance of the method on larger and more complex datasets. Small datasets can limit the ability to generalize the findings, and the method's effectiveness on larger datasets remains uncertain without further testing.

---

> ### Author Response · Authors · 2024-08-14
>
> Thank you so much for your review and letting us know your valuable comments. Please find below our responses to your comments.
>
> **Re Q1:** We would like to address this concern from two perspectives:
>
> - The proposed grSubgraphX framework has significantly reduced computational complexity, from 11 hours per sample (infeasible) to less than 2 minutes per sample (Table 3). While there are still margins for progress, we believe this represents a substantial improvement.
>
> - Additionally, the size of the dataset is not a major concern, as explainability methods are typically used for only a subset of samples in practice. Given this, the computational complexity should not impede the applicability of the model in real use cases and its scalability.
>
>
> **Re Q2:** Please note that there is no such issue with the grXAI framework, as it can handle any graph as long as the pre-trained GNN model can handle it. The grXAI framework is a perturbation-based graph explainability method that identifies important subgraphs for the pre-trained model by perturbing the input graph. Therefore, there is no need for predefined nodes or edge types. We will further clarify this point in the revised version of the manuscript.
>
>
> **Re Q3 and Q4:** Thank you for your comment. We would like to point out that we followed the previous papers for benchmarking and comparison purposes. We have included five real-world datasets for graph embeddings, and one real-world dataset for node embeddings, and only one synthetic dataset in total. These are the main datasets used in the previously published works on graph explainability and graph representation learning  (Wu et al., 2018,  You et al., 2020, Zhang et al., 2023, and Yuan et al., 2020; 2021). Among the five real-world datasets, we further investigated MUTAG results as a case study, for which a deeper understanding of the underlying mechanism connecting structural features to the biological property is available (Qualitative studies in section 5.2).
>
> **Re Q5:** We would like to remark that there is no learning procedure in the grXAI framework. Indeed, post-hoc graph explainability, in general, assumes that there is a pre-trained GNN and the goal of the explainability method is to identify the important subgraph within a specific input graph. As long as one can train a graph representation model using any GNN as a black box and any dataset, grXAI does not have any restrictions. Therefore, the size of dataset is not an issue for the proposed framework.

---

### Review · Reviewer_XfQg · 2024-07-29

**Summary Of Contributions:**

In their paper "Score-based explainability for graph representations", the authors develop an 'explainability' method (grXAI) for GNN-based graph representation learning. This means that given a trained GNN and given an input graph, grXAI identifies a small subgraph that is important for the output representation, in the sense that if the subgraph is removed from the graph, then the output would change a lot. The authors compare their method to a single existing competitor and show that grXAI performs much better.

Disclaimer: I know little about explainability research in general, and nothing about explainability methods for graph learning. Therefore I unfortunately cannot judge on the novelty and on how this work relates to the existing literature.

**Audience:**

Yes

**Claims And Evidence:**

No

**Requested Changes:**

MAJOR COMMENTS

* For me it was difficult to understand sections 3 and 4. Some parts are explained several times with many repetitions. Some parts are not explained at all. Some things that should be explained in the prior work section (section 3) only appear in the methods section (section 4). Please re-think and re-organize sections 3--4 to make them understandable for somebody not very well familiar with the literature.

  * Section 3.1: "Shapley value" -- define, give the formula, explain

  * Section 3.1: "SubgraphX employs MCTS" -- how? explain what it means and how the search is performed

  * Section 3.2: "from a given baseline" -- what does "baseline" mean here?

  * Section 3.2: G* here denotes a subgraph but earlier in Section 3.1 G* meant something else.

  * Section 3.2: What is \Nu_*? It's not defined.

  * Section 4: "we first review the typical setup" -- it would be helpful if all review is contained in Section 3

  * Section 4.1: the entire first paragraph -- shouldn't it appear earlier? You defined Score() here, but you used Score() notation already in Section 3.2.

  * Eq (2): "naive" should be typed in \mathrm{}

  * Section 4.1: "searching for subgraphs can be computationally expensive" -- very unclear. What searching are you talking about? So far the set of subgraphs G^{(i)} are simply given (that's how I understood Section 3.1), so there is no "search".

  * Section 4.1: "averaging multiple connected subgraphs ... may not always be a connected subgraph" -- very unclear. What does "averaging subgraphs" even mean? Equation (1) defines a score for each subgraph. If each subgraph G^{i} is connected, then how would a disconnected subgraph appear?? Super confusing.

  * Section 4.2: first paragraph feels like a repetition

  * End of page 6: the SubgraphX procedure should have been explained in Section 3!

  * End of page 6: "four fundamental axioms" -- very unclear! What axioms, what do they mean, where did they come from?


* Section 5.2: The methods grSubgraphX, grIG, grSaliency were not fully defined. For grSubgraphX there was at least some explanation in Section 4, even though how exactly the search happens was unclear to me. How grIG and grSaliency work, was not explained.

* Section 5.2: The whole experimental setup is unclear. How do you generate results at different values of sparsity? Is sparsity a hyperparameter of your methods? What do the methods even output? Some score value for each subgraph? For each subgraph in what set? For _some_ set of subgraphs? How do you obtain subgraphs for a given sparsity level?

* Figure 4: what do we see in Figure 4, what graph layout is this? Is it GRACE with 2D network output??

* The only comparison in Figure 1 is TAGE. How does TAGE work? I think it should be properly explained in Section 3.


MINOR COMMENTS

* Table 1: very small font, why not make it larger? Also add references (citations) to each row.

* Section 5, first paragraph: "due to space limitations" -- there are no space limitations in TMLR.

* Figure 2: font sizes are way too small, the figure is impossible to read in the print-out.

* Table 2: font can be larger.

**Strengths And Weaknesses:**

Strengths: The methods seems to be simple and builds on prior work in a straightforward way. Experimental comparisons show good performance.

Weaknesses: Many parts of the methodology are unclear and explanations are incomplete. Figures and tables are hard to read. Overall presentation requires more work.

Overall, the paper seems to be a good fit for TMLR and can be accepted after major revision needed to make it understandable.

---

> ### Author Response · Authors · 2024-08-14
> **Part I**
>
> Thank you so much for your constructive comments. We will make sure to revise the paper accordingly. Let us try our best to address your main concerns and clarify a few points while we are working to modify the paper.
>
> **Re MC1:** In the graph explainability methods, we consider that we have a pre-trained GNN model ($f(\cdot): \mathcal{G} \longrightarrow \mathcal{Y}$), and the goal is to identify an important subgraph ($G^*$) within the input graph ($G \in \mathcal{G}$) that is responsible for the GNN output. To achieve this goal, score-based graph explainability methods have two key components to find important subgraphs as explanations: the scoring function and the subgraph extraction component.
>
> Let us assume that we have access to a scoring function that can calculate the importance of each subgraph in some way. Therefore, the model predictions for subgraphs, i.e. $f(G^*)$, are scored against the actual predictions, $f(G)$, using the scoring function. For example, the scoring function is the Shapley value in the case of SubgraphX.
>
> A straightforward way to obtain $G^*$ is to enumerate all possible subgraphs in $G$, calculate their importance through the scoring function, and select the most important one as the explanation. However, such a brute-force approach is an intractable combinatorial problem, and the potential candidates increase exponentially with the number of nodes.
>
> To solve this issue, we can either incorporate search algorithms to explore the subgraphs efficiently, as done in SubgraphX, or calculate the importance of each node independently, similar to the gradient-based methods.
>
> In the case of SubgraphX, the scoring component is the Shapley value (Kuhn & Tucker, 1953), and the subgraph extraction component is Monte Carlo Tree Search (MCTS) (Silver et al., 2017; Jin et al., 2020). MCTS builds a search tree in which the root is associated with the input graph and each of the other nodes corresponds to a connected subgraph. Each edge in the search tree denotes that the graph associated with a child node can be obtained by performing node-pruning from the graph associated with its parent node.
>
> In the case of gradient-based methods, they don’t need a subgraph extraction component. Intuitively, they consider the gradient of the GNN prediction concerning each node, representing how sensitive the GNN prediction is to that node. Then, they sort the nodes based on their importance values and select the top K ones as the explanation of the input graph.
>
> - **Re** Section 3.2: "from a given baseline" -- what does "baseline" mean here?
>
> >Integrated Gradients calculates the integral of gradients of the model's prediction with respect to the input features along a linear path from the baseline to the input. The baseline provides a comparison point to understand each feature's influence. By starting from the baseline and moving toward the input, IG captures the contribution of each feature to the model's prediction. Common baselines include a zero vector, mean or median input, and random or perturbed inputs.
>
> - **Re** Section 3.2: G* here denotes a subgraph but earlier in Section 3.1 G* meant something else.
>
> >$G^*$ always refers to the important subgraph identified as the explanation for the input graph, which can be a connected subgraph or a set of disconnected nodes.
>
> - **Re** Section 3.2: What is \Nu_*?
>
> >Since the gradient-based method only identifies important nodes, the important subgraph to explain would be the set of important nodes, and $\mathcal{V}_*$ refers to the total number of important nodes.
>
> - **Re** Section 4: "we first review the typical setup" -- it would be helpful if all review is contained in Section 3
>
> >We are modifying Sections 3 and 4 as you instructed. However, please note that Section 4 addresses multi-dimensional output, whereas Section 3 outlines explainability for the supervised setting for the particular label of interest. We will ensure that this distinction is clear in the revised version.
>
> - **Re** Eq (2): "naive" should be typed in \mathrm{}
>
> >Thanks for pointing out the typo. We have fixed it in the revised version.
>
> - **Re** Section 4.2: first paragraph feels like a repetition
>
> > Please note that there is a difference between the notation of GNN in the supervised setting in Section 3 and the unsupervised setting in Section 4.
>
> - **Re** End of page 6: the SubgraphX procedure should have been explained in Section 3!
>
> >This is not SubgraphX, rather as stated in the title of the paragraph, it is the extension of SubgraphX for graph representation explainability using the proposed grXAI framework.

---

> > ### Author Response · Authors · 2024-08-14
> > **Part II**
> >
> > - **Re** Section 4.1: "averaging multiple connected subgraphs ... may not always be a connected subgraph" -- very unclear.
> >
> > >The main challenge of the available score-based methods explained in Section 3 is that they can only calculate the scoring function for one dimension of the GNN output, $f_j(\cdot)$. In the case of graph representations, the number of latent dimensions of GNN embedding is $d_h$. Therefore, we will have $d_h$ different subgraphs for every dimension of the representation, identified through the subgraph extraction component. To come up with a single important subgraph, we need to combine all the identified subgraphs for every dimension. However, if the different identified subgraphs don’t share nodes, we might end up with multiple subgraphs that are not connected. In practice, this means that even using a method that returns a connected subgraph for each output dimension, there is no guarantee that their combination will also be a connected subgraph.
> >
> >
> > - **Re** End of page 6: "four fundamental axioms" -- very unclear! What axioms, what do they mean, where did they come from?
> >
> > >These desirable axioms introduced by Lundberg & Lee (2017) can guarantee the correctness and fairness of the explanations. We will revise the paper to better reference them.
> >
> > **Re MC2:** We will add a formal description of grIG and grSaliency at the end of section 4.
> >
> > **Re MC3:** Similar to other post-hoc graph explainability methods, the grXAI framework takes a pre-trained model $f$ and the graph $G$ as inputs and provides a single subgraph $G^*$ as output. Specifically, the model searches for potential subgraphs in $G$ whose size is less than $N_{\text{min}}$​ (where $N_{\text{min}}$​ in equation (1) acts as a hyperparameter to control sparsity), calculates their importance scores, and selects the subgraph with the highest value. Please note that this is the standard approach in post-hoc graph explainability methods and we followed previous literature. We will be sure to clarify this in the revised version as well.
> >
> > **Re MC4:** Figure 4 shows the input graph structure. Since the task is node embedding and there is only one large graph, we show only the L-hop neighboring nodes of the target node that we are trying to explain for ease of visualization.
> >
> > **Re MC5:** Thanks for your comment. We will add more background about TAGE in the revised version of the manuscript.

---

> ### Comment · Reviewer_XfQg · 2024-08-26
>
> Thanks for your responses, and apologies for not being able to reply earlier.
>
> I have only skimmed your edits now (it would be easier if you could upload a latexdiff pdf highlighting the changes that you did in revision), but here is one thing that I noticed is still confusing to me:
>
> > > Section 4.1: "averaging multiple connected subgraphs ... may not always be a connected subgraph" -- very unclear.
> >
> > The main challenge of the available score-based methods explained in Section 3 is that they can only calculate the scoring function for one dimension of the GNN output. [...] we will have different subgraphs for every dimension of the representation [...] To come up with a single important subgraph, we need to combine all the identified subgraphs for every dimension. However, if the different identified subgraphs don’t share nodes, we might end up with multiple subgraphs that are not connected.
>
> Sorry, I still don't understand what "averaging" means here. In Section 4.1 there is formula (2) for the "naive" score, obtained with averaging importance scores over all dimensions. If we apply this naive score formula to each possible connected subgraph, and identify subgraph with the highest naive score, then it will clearly be connected.
>
> In your text below the formula and also in your response to me here in the rebuttal you seem to be referring to some _other_ procedure where the most important subgraph for each component is found, and then these subgraphs are somehow "averaged" (this "averaging" is never defined). But this is not what happens in Equation (2), which makes the argument very confusing.
>
> Note that this is a very important argument for the paper, as the paper presents the issue of potentially disconnected subgraphs as the major limitation of this naive approach ("connected"ness is even mentioned in the abstract).

---

> > ### Author Response · Authors · 2024-08-26
> >
> > Thank you for your feedback. We appreciate the opportunity to clarify this point. To address your concern, we are indeed referring to Equation (2) in Section 4.1. Let’s consider a scenario where the output dimension is 2 and $f_1(G) = \alpha$. In this case, we can rewrite Equation (2) as follows: $\overline{\mbox{Score}}_{\mbox{Naive}}(f(\cdot), G, G^{(i)}) = $
> >
> > $\sum_{k=1}^{d_y}  f_k(G) \cdot \mbox{Score}(f_k(\cdot), G, G^{(i_j)}) = \alpha \[ \mbox{Score}(f_1(\cdot), G, G^{(i_1)})] + (1-\alpha) \[\mbox{Score}(f_2(\cdot), G, G^{(i_2)})\]$
> >
> > It is important to note that, according to Equation (1), only the output of each score function, $\mbox{Score}(f_k(\cdot), G, G^{(i_j)})$, is a connected subgraph. Therefore, we obtain two connected subgraphs, $G^{(i_1)}$ and $G^{(i_2)}$, identified for $f_1(G)$ and $f_2(G)$ respectively. In this context, $G^{(i)}$, which represents the most important subgraph in Equation (2), is a combination of these two subgraphs, $G^{(i_1)}$ and $G^{(i_2)}$.
> > This combined subgraph $G^{(i)}$  will be connected if and only if $G^{(i_1)}$ and $G^{(i_2)}$ are connected to each other (i.e., if they share at least one node).
> >
> > To illustrate this with an example, consider a graph structured as 1—2—3—4—5, where the numbers are nodes. Possible connected subgraphs for $G^{(i_1)}$ and $G^{(i_2)}$ might be 1—2 and 4—5, respectively. In this scenario, one potential $G^{(i)}$ would be $\[1—2, 4—5\]$, which is not a connected subgraph (even if the individual subgraphs are connected).
> >
> > We hope this explanation clarifies the points of confusion. Based on your feedback, we have also changed $G^{(i)}$ on the right-hand side of Equation (2) with $G^{(i_j)}$ to make equation (2) more clear.

---

> > > ### Comment · Reviewer_XfQg · 2024-08-27
> > > **Still confused**
> > >
> > > I appreciate your response but I remain confused.
> > >
> > > You wrote "only the output of each score function ... is a connected subgraph". But the "output" of any score function is just a number (for each input subgraph). What I guess you mean is that the argmax of each score function is a connected subgraph, simply because the argmax in Equation 1 goes over all connected subgraphs. Right? Okay, so far so good.
> > >
> > > Now we get to Equation 2, which defined a new "naive" score function. This is a function that maps every possible subgraph $G^{(i)}$ to a number. Great. Now we can define $G^*$ as in Equation 1 to be the argmax over all connected subgraphs of this naive score function. OBVIOUSLY this argmax $G^*_\mathrm{naive}$ is going to be connected.
> > >
> > > Instead, what you seem to be saying, is that score function for component 1 has some $G^*_1$ as argmax, while score function for component 2 has some $G^*_2$ as argmax, and these $G^*_1$ and $G^*_2$ may be non-overlapping. I understand that statement. But why do we care?? You defined a scoring function in Equation 2, and you defined the argmax procedure in Equation 1, so the argmax of Score_naive is going to be connected!
> > >
> > > Also, your new Equation 2 does not make any sense to me. What is $G^{(i_j)}$ on the right side? What is $j$? It does not appear in the formula. Did you mean $k$ instead of $j$? But then what exactly is $G^{(i_k)}$ supposed to mean? Do you mean some argmax, so some $G^*$? As written, none of that make sense to me, sorry.

---

> > > > ### Author Response · Authors · 2024-08-28
> > > >
> > > > Thank you for your feedback and thoughtful review. We are aligned on the first part of your discussion. We were referring indeed to the subgraph with the highest importance score. However, it's important to clarify that Equation 1 does not iterate over all possible subgraphs. Instead, as we mentioned in section 3.2, it utilizes Monte Carlo Tree Search (MCTS) to guide the search process. The task of exploring all possible connected subgraphs is a combinatorial problem, with the number of potential candidates increasing exponentially. This makes a full search not only infeasible but also intractable from a computational standpoint.
> > > >
> > > > Regarding Equation 2, while your interpretation is theoretically accurate—that we could, in theory, examine all subgraphs, compute the importance score based on the naive score in Equation 2, and select the one with the highest score—this approach is computationally impractical as we mentioned above.
> > > >
> > > > To manage this computational challenge, we adopted MCTS, similar to the method used in SubgraphX, to efficiently identify important subgraphs for each output component separately, and then merged them. We referred to this as the Naive Score function because it simply represents a basic way of identifying the most important subgraphs using the currently available implementation of SubgraphX. This method achieves a computational complexity of $O(d_y . \alpha . N_{min})$, where $N_{min}$ controls the desired sparsity of the subgraph, $d_y$ is the output dimension, and $\alpha$ represents the complexity of computing the score function.
> > > >
> > > > Even with this strategy, computational complexity remains a challenge. Using a small $N_{min}$ and combining multiple smaller subgraphs to create a larger subgraph allows for handling only a few examples. For instance, using this naive approach for a single graph required 11 hours of computation, as we reported in Table 3, whereas our proposed method (i.e., Equation 6) reduced the runtime to less than 2 minutes.
> > > >
> > > > While one could also use the naive score function with MCTS to identify a single connected subgraph, as you suggested, this would require a new implementation of SubgraphX, as the current version does not accept a custom scoring function as input. Furthermore, even with this new implementation, the computational complexity would be $O(d_y . \alpha . N^*_{min})$, where $N^*_{min}$ must be set larger than $N_{min}$ to achieve the same level of sparsity. This makes the computational complexity even more challenging and not scalable in practice for moderately sized graphs.
> > > >
> > > >
> > > > A couple of points to consider: (i) Neither the naive approach we adopted nor the one you suggested (using the naive score function and argmax over this) is practical due to the computational complexity being a major bottleneck. The grXAI framework is designed primarily to address this issue, regardless of the score function used, whether it’s IG/ Saliency (which cannot handle connected subgraphs), or SubgraphX (which can provide connected subgraphs). (ii) We will revise the paper to emphasize that obtaining connected subgraphs through the naive scoring function, as you suggested, is indeed theoretically possible. (iii) Nevertheless, we still argue that the proposed method in Equation 6, grSubgraphX, is currently the only method that can practically (i.e., in a computationally efficient way) provide connected subgraphs for graph representation. In contrast, using the naive score function alone is not feasible in practice.
> > > >
> > > > Lastly, thank you for pointing out the typo. We have corrected the notation from j to k in Equation 2 to align with the rest of the notation used.

---

> ### Comment · Reviewer_XfQg · 2024-08-28
>
> What you wrote here makes sense. However, *none of that is explained in the paper text*! The current paper text simply gives Equation 2, which defines a naive score function, so the natural interpretation is that you are going to use some approach (e.g. MCTS) to search for argmax, as stated earlier in Equation 1. Then a few lines below you say "Another limitation of averaging multiple connected subgraphs" -- but the text has NEVER mentioned "averaging" connected subgraphs before! This needs to be explained, as you did here.
>
> Also, as currently written, Eq 2 is unclear because $G^{(i_k)}$ is undefined. To be honest, as currently written, Eq 2 does not make any sense to me. The left-hand side has G^i as input. But the right-hand side does not have G^i. The formula does not work like that.

---

> > ### Author Response · Authors · 2024-08-29
> >
> > Thank you for your constructive feedback. Our initial intention was to keep Section 4.1 focused on theoretical discussion, using Equation 2 as a general description for all score-based approaches without delving into specific implementation details. However, we understand now that this approach may have caused confusion and made that part difficult to follow. We appreciate your insights, and we have revised the section to incorporate these points,  providing a clearer explanation of the averaging of multiple connected subgraphs, as well as how it relates to the score function. We have also revised Equation 2 to ensure it is properly defined and coherent. Specifically, we have reverted $G^{(i_j)}$ to the original version $G^{(i)}$, and following that equation clarified how we handle SubgraphX. Your comments have been instrumental in highlighting areas for improvement, and this revision aims to address the issues you raised and improve the overall clarity of the paper.

---

> > > ### Comment · Reviewer_XfQg · 2024-08-29
> > >
> > > Thanks for the edits. Now the text is clear.
> > >
> > > Please note that Figure 4 caption layout is currently broken.

---

> > > > ### Author Response · Authors · 2024-08-29
> > > >
> > > > Thank you for taking the time to review our paper and provide constructive feedback.

---

### Decision · Action_Editor_Xiw9 · 2024-09-09

**Recommendation:** Accept as is

**Comment:**

This paper extended prior work subGraphX into explaining unsupervised graph neural networks. Reviewers generally find the paper provides interesting insights and is solving a new problem, albeit the methodological contribution is minor. After discussion with the reviewers, new experiments are added which improved the comprehensiveness of the paper. 2 reviewers are leaning accept and 1 reviewer is leaning reject, but there is no substantial issues that are not resolved in the final version of the paper other than the minor methodological contribution. Given the TMLR review criteria, the editor decides to accept the paper as it provides new insights to the task of explaining unsupervised graph neural networks.

**Audience:**

Audience on explainability and audience on graph neural networks should be interested in this paper.

**Claims And Evidence:**

The paper extended prior work SubGraphX into explaining unsupervised graph neural networks. Experiments against multiple baselines on multiple datasets showed improvements over prior work.